# Network-based approach to prediction and population-based validation of in silico drug repurposing

Feixiong Cheng[1,2], Rishi J. Desai [3], Diane E. Handy[4], Ruisheng Wang[4], Sebastian Schneeweiss[3], Albert-László Barabási[1,2,5,6] & Joseph Loscalzo[4]

Here we identify hundreds of new drug-disease associations for over 900 FDA-approved drugs by quantifying the network proximity of disease genes and drug targets in the human (protein–protein) interactome. We select four network-predicted associations to test their causal relationship using large healthcare databases with over 220 million patients and state-of-the-art pharmacoepidemiologic analyses. Using propensity score matching, two of four network-based predictions are validated in patient-level data: carbamazepine is associated with an increased risk of coronary artery disease (CAD) [hazard ratio (HR) 1.56, 95% confidence interval (CI) 1.12–2.18], and hydroxychloroquine is associated with a decreased risk of CAD (HR 0.76, 95% CI 0.59–0.97). In vitro experiments show that hydroxychloroquine attenuates pro-inflammatory cytokine-mediated activation in human aortic endothelial cells, supporting mechanistically its potential beneficial effect in CAD. In summary, we demonstrate that a unique integration of protein-protein interaction network proximity and large-scale patient-level longitudinal data complemented by mechanistic in vitro studies can facilitate drug repurposing.

[1] Center for Complex Networks Research and Department of Physics, Northeastern University, Boston, MA 02115, USA. [2] Center for Cancer Systems Biology and Department of Cancer Biology, Dana-Farber Cancer Institute, Boston, MA 02215, USA. [3] Division of Pharmacoepidemiology and Pharmacoeconomics, Department of Medicine, Brigham and Women's Hospital, Harvard Medical School, Boston, MA 02115, USA. [4] Department of Medicine, Brigham and Women's Hospital, Harvard Medical School, Boston, MA 02115, USA. [5] Channing Division of Network Medicine, Department of Medicine, Brigham and Women's Hospital, Harvard Medical School, Boston, MA 02115, USA. [6] Center for Network Science, Central European University, Budapest 1051, Hungary. These authors contributed equally: Feixiong Cheng, Rishi J. Desai  Correspondence and requests for materials should be addressed to J.L. (email: jloscalzo@rics.bwh.harvard.edu)

Although investment in biomedical and pharmaceutical research and development has increased significantly over the past 20 years, the annual number of new treatments approved by the US Food and Drug Administration (FDA) has not significantly increased[1]. Among the reasons for this shortcoming in contemporary drug development are a lack of well-established predictive pharmacokinetics/pharmacodynamics approaches, and concerning safety and tolerability profiles for new chemical entities from preclinical studies to clinical trials[2]. In addition to these well recognized explanations, another important factor limiting more effective drug development may be continued adherence to the classical (one gene, one drug, one disease) hypothesis. Focusing on just single targets results in failure to anticipate off-target toxicity, unintended beneficial effects, or multiple target interactions leading to suboptimal efficacy[3,4]. Without full knowledge of the broader network context of the molecular determinants of disease and drug targets in the protein–protein interaction network (human interactome), investigators cannot develop meaningful approaches for efficacious treatment of complex diseases[5].

Novel approaches, such as network-based drug-disease proximity, that shed light on the relationship between drugs (drug targets) and diseases [molecular (protein) determinants in disease modules][6–8] can serve as a useful tool for efficient screening of potentially new indications for approved drugs with well-established pharmacokinetics/pharmacodynamics, safety and tolerability profiles, or previously unidentified adverse events[9–12]. However, in order to prioritize the repurposed candidates or suggest novel interventions based on drug-disease associations identified by network-based approaches, rigorous validation is mandatory. Since network-based drug repurposing focuses on drugs that are already approved and are used in clinical practice, such hypothesis testing is possible using large-scale patient-level data collected during routine healthcare. Such data are regularly used to generate actionable evidence regarding effectiveness, harm, use, and value of medications to supplement evidence generated in randomized controlled trials; these trials that lead to drug approval are typically limited in scope owing to a relatively modest study sample size, comparatively short follow-up time, and frequent underrepresentation of the most relevant populations[13]. The unique strengths of routine healthcare data that make them ideal for validating hypotheses generated by network-based predictions include their provision of large patient populations useful for detecting small differences, and the availability of a large number of patient factors recorded without any recall bias, including demographics, comorbid conditions, and medication use, that allow for high-dimensional covariate adjustment to minimize confounding[14–16].

In this study, we developed a systems pharmacology-based platform that quantifies the interplay between disease proteins and drug targets in the human protein–protein interactome with state-of-the-art pharmacoepidemiologic methods for hypothesis validation using longitudinal data with over 220 million patients. We followed this analysis with in vitro assays to test potential drug mechanisms. As proof of the utility of the overall approach, we focused on cardiovascular (CV) outcomes given their high prevalence in the population, as an exemplary set of diseases with which to identify associations between drugs used for non-cardiac indications and CV outcomes. We demonstrate that an integrated approach incorporating network proximity together with large-scale patient longitudinal data and in vitro experimental assays offers an effective platform by which to identify and validate novel associations that can be used to minimize unanticipated adverse drug effects and optimize drug repurposing. These results suggest that this integrative approach can be generalized to other drugs/disease combinations.

## Results

**An atlas of drug effects via network proximity.** Our previous studies have demonstrated that disease gene products (proteins) are likely to cluster in the same network neighborhood or disease module within the human protein–protein interactome[6,17,18]. Drug targets representing nodes within molecular networks are often intrinsically coupled in both therapeutic and adverse effects. We, therefore, proposed that for a drug with multiple targets to be on-target effective for a disease or to cause off-target adverse effects (Supplementary Fig. 1a), its target proteins should be within or in the immediate vicinity of the corresponding disease module in the human interactome[8,10]. We chose CV diseases as a test case of this principle due to their prevalence in the population and their high morbidity and mortality. To examine drug effects on CV diseases, we used a network proximity measure that quantifies the relationship between CV-specific disease modules and drug targets in the human protein–protein interaction (PPI) network (Supplementary Fig. 1b). To improve the data quality of the human interactome, we used only five types of experimental data: (a) binary PPIs obtained using systematic, unbiased, high-throughput yeast-two-hybrid (Y2H) systems[19]; (b) kinase-substrate interactions from literature-derived low-throughput and high-throughput experiments; (c) binary PPIs from three-dimensional (3D) protein structures; (d) signaling networks from literature-derived low-throughput experiments; and (e) literature-curated PPIs identified by affinity purification followed by mass spectrometry (AP-MS), Y2H, and/or literature-derived low-throughput experiments in which every interaction is supported by multiple sources of experimental evidence (Methods section). The updated human interactome defined in this way includes 243,603 PPIs connecting 16,677 unique proteins (Supplementary Data 1). We also compiled 984 FDA-approved drugs by pooling the reported experimental drug-target binding affinity data: median effective concentration ($EC_{50}$), median inhibitory concentration ($IC_{50}$), inhibition constant/potency ($K_i$), or dissociation constant ($K_d$), each ≤10 micromolar (μM) as a cutoff. We first calculated a $z$-score $\left( z = \frac{d-\mu}{\sigma} \right)$ for quantifying the significance of the shortest path lengths $d(s,t)$ between targets ($t$) of a drug ($T$) and proteins ($s$) associated with the CV module ($S$) where the closest distance between a drug and a disease $d(S,T)$ is defined as:

$$d(S, T) = \frac{1}{\|T\|} \sum_{t \in T} \min_{s \in S} d(s, t). \qquad (1)$$

We constructed the reference distance distribution corresponding to the expected network topological distance between two randomly selected groups of proteins matched to size and degree (connectivity) as the original disease proteins and drug targets in the human interactome (cf. Methods). The $z$-score reduces the study bias (e.g., hub nodes or those nodes with high connectivity) in the shortest-path methods as described in our previous study[10]. In total, we computationally investigated 984 FDA-approved drugs [177 CV drugs and 807 non-CV drugs defined by first-level Anatomical Therapeutic Chemical (ATC) classification codes] and 23 types of CV outcomes (specific CV diseases) (Supplementary Table 1). Relying on 177 FDA-approved CV drugs and their known CV indications, we found that the area under the receiver operating characteristic curve (AUC) is over 70% using the network proximity measure (Supplementary Fig. 2), revealing high accuracy for identifying the well-known drug-disease relationships. In addition, we compared the network proximity measure, closest ($z$-score), against three other network distance-based measures between drug targets and the disease module[10]:

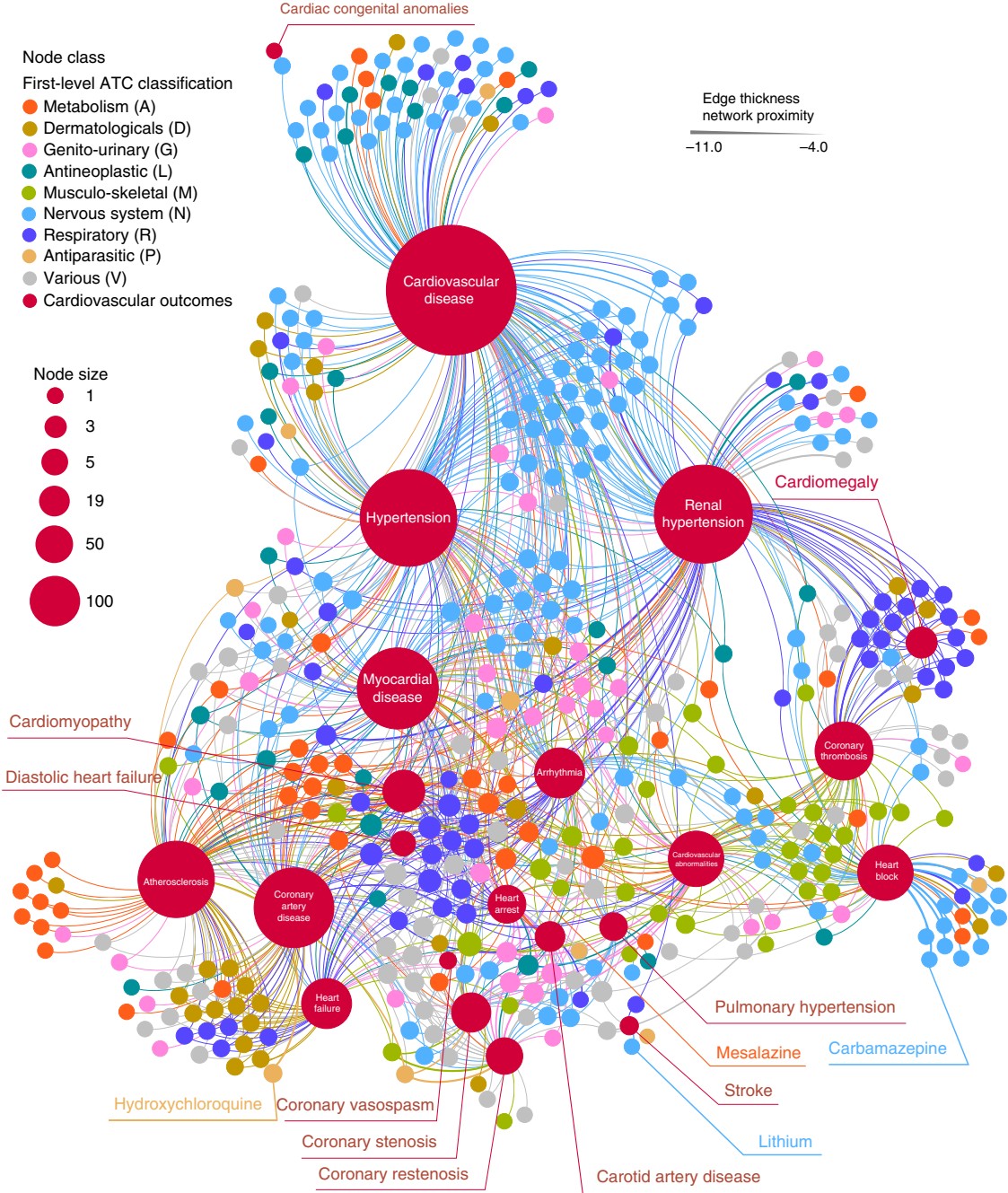

**Fig. 1** The predicted drug-disease network. The high-confidence predicted drug-disease association network connects 22 types of cardiovascular disease (outcomes) (red circles) and 431 FDA-approved non-cardiac drugs. The edges between drugs and diseases are weighted and highlighted by different color representing the calculated $z$-score (Supplementary Data 2 and Methods section). Four selected drug-disease pairs, including carbamazepine-coronary artery disease (CAD) with $z = -2.36$, hydroxychloroquine-CAD ($z = -3.85$), mesalamine-CAD ($z = -6.10$), and lithium-stroke ($z = -5.97$), tested in patient data (Figs. 2 and 3), are highlighted. Drugs are colored by the first-level anatomical therapeutic chemical (ATC) classification system codes. The node size scales indicate the degree (connectivity) of nodes in the network

(1) shortest, (2) kernel, and (3) centre. We found that the closest distance-based $z$-score outperformed all three alternative network distance measures (Supplementary Fig. 3). We, therefore, used the closest distance-based $z$-score in the follow-up studies. Figure 1 illustrates the high-confidence predicted drug-CV disease associations ($z < -4.0$) connecting 431 non-CV drugs to 22 specific CV disease modules. We next proposed that this atlas of the predicted associations between non-CV drugs and CV disorders offers a useful resource with which to prioritize new CV

indications or highlight potential (unexpected) adverse cardiac events for various approved drugs.

**Validating possible causal associations in patient data**. We selected four target associations between non-CV drugs and CV diseases identified by the network proximity measure (closest) for hypothesis validation by analyzing over 220 million patients in healthcare databases (Fig. 2). Target associations were further

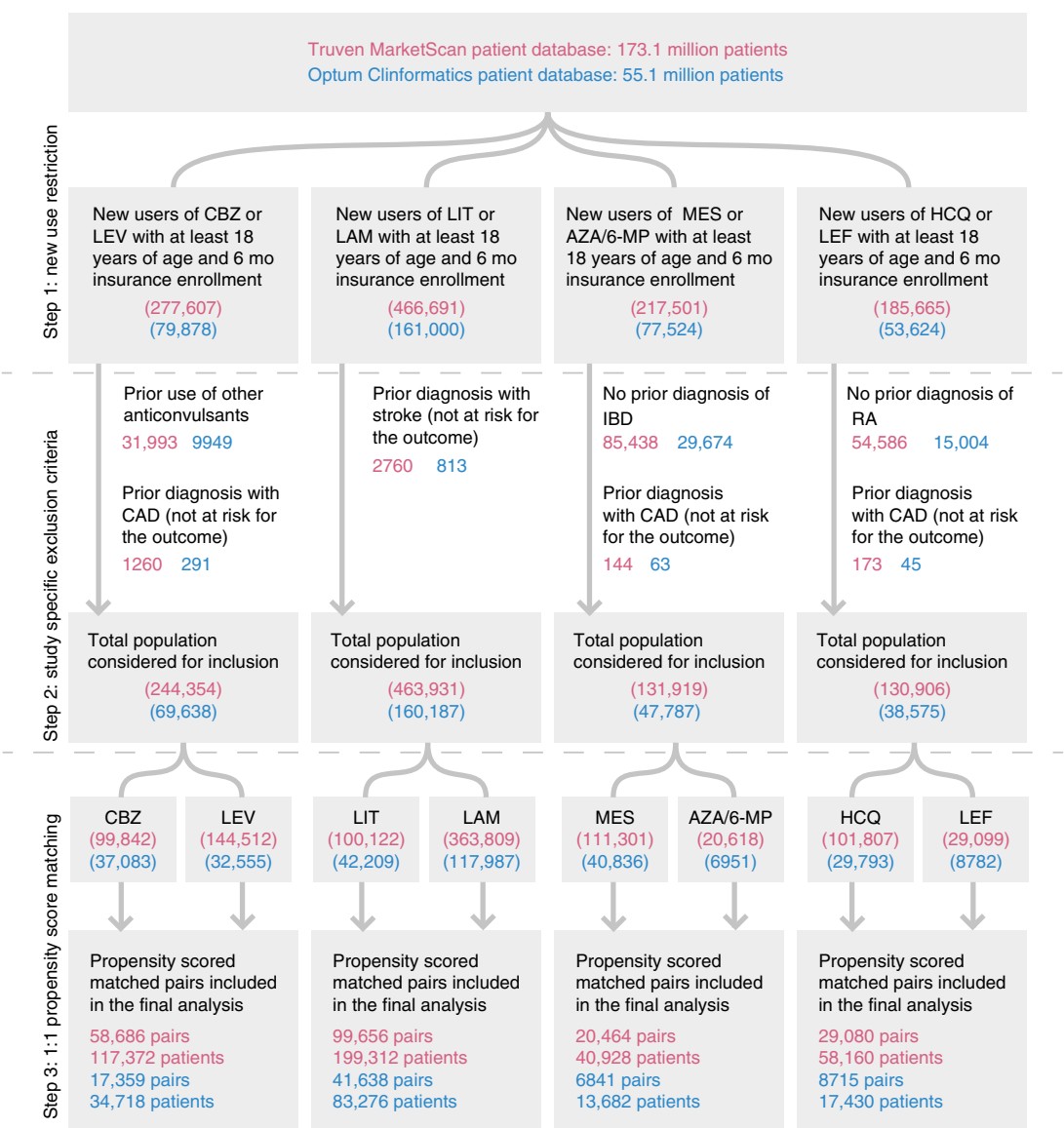

**Fig. 2** Flow-chart of the pharmacoepidemiologic investigations using Truven MarketScan and Optum Clinformatics patient databases. AZA azathioprine, 6-MP 6-mercaptopurine, CBZ carbamazepine, HCQ hydroxychloroquine, IBD inflammatory bowel disease, LAM lamotrigine, LEF leflunomide, LEV levetiracetam, LIT lithium, MES mesalamine, RA rheumatoid arthritis. The balance achieved in patient characteristics and outcome risk factors between the two treatment groups compared through 1:1 PS-matching are provided in Supplementary Tables 2–5

selected using subject matter expertise based on a combination of factors: (i) strength of the network-based predicted associations (a higher network proximity score in Supplementary Data 2); (ii) novelty of the predicted associations through exclusion of known adverse CV events of non-CV drugs (Methods section); (iii) availability of sufficient patient data for meaningful evaluation (exclusion of infrequently used medications); (iv) availability of an appropriate comparator treatment that is used for the same underlying (non-CV) indication as the drug of interest and predicted to have no association with the intended CV diseases via network proximity analysis (defined reference groups or negative controls); and (v) the fidelity with which the predicted CV diseases were recorded in insurance claims databases. Applying these criteria resulted in four network-based predictions: (1) carbamazepine ($z = -2.36$) vs. levetiracetam (comparator control, $z = -0.07$), drugs normally used to treat epilepsy, with CAD; (2) mesalamine ($z = -6.10$) vs. azathioprine (comparator control, $z = -0.09$), drugs normally used to treat inflammatory bowel

disease, with CAD; (3) hydroxychloroquine ($z = -3.85$) vs. leflunomide (comparator control, $z = -1.87$), drugs normally used to treat rheumatoid arthritis, with CAD; and (4) lithium ($z = -5.97$) vs. lamotrigine (comparator control, $z = 0.19$), drugs normally used to treat bipolar disorder, with stroke.

Using two large US-based commercial health insurance claims databases connected with the validated Aetion evidence platform[20], we next conducted four cohort studies to evaluate the predicted associations based on individual level longitudinal patient data and pharmacoepidemiologic methods, including a new-user active comparator design, propensity score (PS) adjustment for confounding, and multiple sensitivity analyses[21]. Figure 2 summarizes the total patients included in the four cohorts along with specific reasons for exclusion in the Truven MarketScan and Optum Clinformatics databases. Overall, based on more than 50 covariates included in the PS, we included: (1) 76,045 carbamazepine initiators matched 1:1 to 76,045 levetiracetam initiators; (2) 27,305 mesalamine initiators matched

**Table 1 Summary of sample sizes, follow-up time, events, and incidence rates in pharmacoepidemiologic investigations**

| Parameter | Study 1 | | Study 2 | | Study 3 | | Study 4 | |
|---|---|---|---|---|---|---|---|---|
| | New users of carbamazepine | New users of levetiracetam | New users of lithium | New users of lamotrigine | New users of mesalamine | New users of azathioprine/6-MP | New users of hydroxychloroquine | New users of leflunomide |
| Database 1: Truven MarketScan | | | | | | | | |
| Patients | 58,686 | 58,686 | 99,656 | 99,656 | 20,464 | 20,464 | 29,080 | 29,080 |
| Person-years | 27,050 | 31,826 | 46,278 | 61,970 | 8714 | 11,562 | 18,650 | 17,293 |
| Outcomes[a] | 124 | 105 | 67 | 90 | 17 | 27 | 88 | 106 |
| Incidence rate per 100 person-years (95% CI) | 0.46 (0.38, 0.55) | 0.33 (0.27, 0.40) | 0.14 (0.11, 0.18) | 0.15 (0.12, 0.18) | 0.20 (0.11, 0.31) | 0.23 (0.15, 0.34) | 0.47 (0.38, 0.58) | 0.61 (0.50, 0.74) |
| Database 2: Optum Clinformatics | | | | | | | | |
| Patients | 17,359 | 17,359 | 41,638 | 41,638 | 6841 | 6841 | 8715 | 8715 |
| Person-years | 8662 | 8945 | 18,925 | 24,627 | 2589 | 3587 | 5376 | 5043 |
| Outcomes[a] | 56 | 30 | 10 | 31 | 15 | 11 | 29 | 39 |
| Incidence rate per 100 person-years (95% CI) | 0.65 (0.49, 0.84) | 0.34 (0.23, 0.48) | 0.05 (0.03, 0.10) | 0.13 (0.09, 0.18) | 0.58 (0.32, 0.96) | 0.31 (0.15, 0.55) | 0.54 (0.36, 0.77) | 0.77 (0.55, 1.06) |

[a]The outcome is coronary artery disease for Studies 1, 3, and 4; and stroke for Study 2.

1:1 to 27,305 azathioprine initiators; (3) 37,795 hydroxychloroquine initiators matched 1:1 to 37,795 leflunomide initiators; and (4) 141,294 lithium initiators matched 1:1 to 141,294 lamotrigine initiators. Supplementary Tables S2–S5 demonstrate the balance achieved in patient characteristics and outcome risk factors between the two treatment groups compared via 1:1 PS-matching. Table 1 shows the total number of person-years of follow-up, total event counts (incident diseases) in the patient groups, and incidence rates per 1000 person-years for the diseases of interest (95% confidence interval [CI]) for each of the four comparisons of interest stratified by data source.

Figure 3 summarizes the results after pooling the two patient databases for each of the four comparisons before and after PS-matching. In the primary analytical approach of censoring patient follow-up time at discontinuation of the initial treatment ("as-treated" approach), we observed that carbamazepine was associated with a 56% increased risk [hazard ratio (HR) 1.56, 95% confidence interval (CI) 1.12–2.18] of CAD compared with levetiracetam (Fig. 3a), and hydroxychloroquine (Fig. 3d) was associated with a 24% reduced risk of CAD compared to leflunomide (HR 0.76, 95% CI 0.59–0.97). Varying the follow-up assumptions used in the following ways—(1) excluding the first 60 days of follow-up to reduce residual baseline confounding, (2) truncating the follow-up to 1-year to minimize time-varying confounding, and (3) continuing the follow-up for 1-year regardless of treatment discontinuation under an intent-to-treat (ITT) principle–resulted in estimates that were consistent with the primary approach for both the carbamazepine (Fig. 3a) and hydroxychloroquine analyses (Fig. 3d). Mesalamine vs. azathioprine (HR 1.15, 95% CI 0.55–2.42) and lithium vs. lamotrigine (HR 0.71, 95% CI 0.31–1.60) were not consistently associated differentially with the risk of CAD or stroke (Fig. 3b, c and Supplementary Figs. 4 and 5). Therefore, two of the four predicted associations were validated by the large-scale patient data to either decrease the risk of CAD (hydroxychloroquine) or increase the risk of CAD (carbamazepine), supporting our network-based prediction.

**In vitro assay of hydroxychloroquine's mechanism-of-action.** Figure 3d reveals that hydroxychloroquine is associated with a 24% reduced risk of CAD compared to leflunomide (HR 0.76, 95% CI 0.59–0.97). [These very robust data are in agreement with a recent study showing that hydroxychloroquine decreases the incidence of CV events in a small cohort of rheumatoid arthritis patients[22].] Hydroxychloroquine has been approved for the treatment of malaria and rheumatoid arthritis for many years; however, only recently have studies provided relevant mechanistic insights. Hydroxychloroquine accumulates intracellularly in the endosomal/lysosomal compartment where its inhibitory effects on Toll-like receptors 7 and 9 (TLR7 and TLR9) suppress inflammatory responses[23]. We integrated drug targets and disease proteins into the blood vessel-specific protein–protein interaction network (cf. Methods) to identify the overlapping pathways between hydroxychloroquine targets and CAD proteins (Fig. 4a). Two potential pathways were inferred to be involved in the protective effect of hydroxychloroquine in CAD: (a) hydroxychloroquine may activate ERK5 (encoded by *MAPK7*) to prevent endothelial inflammation via inhibition of cell adhesion molecule expression[24]; and (b) hydroxychloroquine may inhibit endosomal activation of NADPH oxidase in response to pro-inflammatory agonists (TNF-α and IL-1β) and may decrease production of pro-inflammatory cytokines in stimulated immune cells[25]. Notably, adhesion molecules (ICAM-1 and VCAM-1)[26] and pro-inflammatory cytokines[27] play essential roles in CAD. Furthermore, a recent meta-analysis has shown

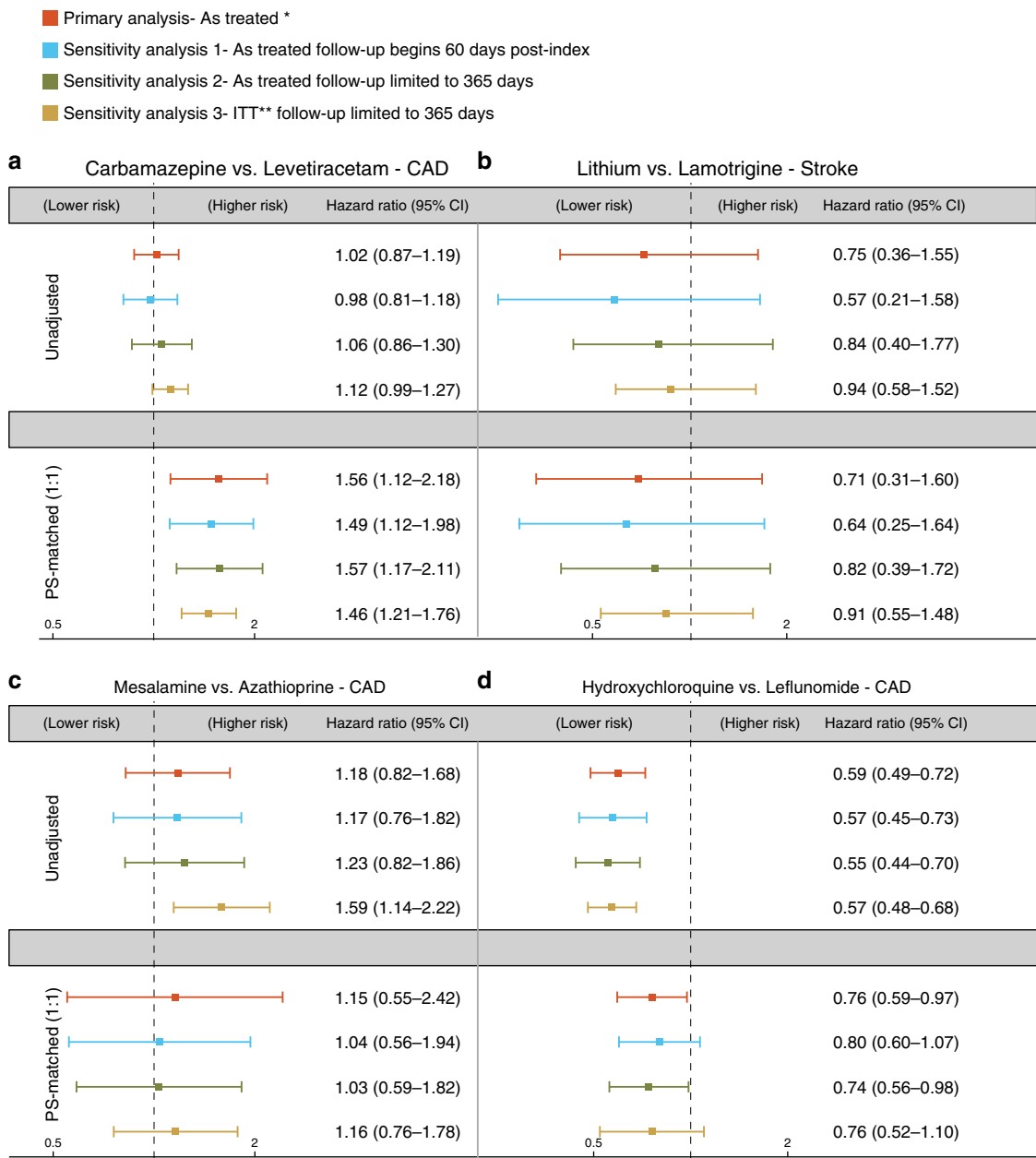

**Fig. 3** Hazard ratios and 95% confidence intervals for four cohort studies. Four cohort studies were performed using the pooled data from four drug pairs (**a-d**) Truven MarketScan and Optum Clinformatics databases (Methods section). In the primary analysis approach (as-treated), follow-up was stopped upon discontinuation of the index medication. Follow-up assumptions were varied in three sensitivity analyses to: (1) exclude the first 60 days of follow-up to reduce unmeasured baseline confounding, (2) truncate the follow-up to 1-year to minimize time-varying confounding, and (3) continue the follow-up for 1-year regardless of treatment discontinuation under an intent-to-treat (ITT) principle. Propensity score (PS) matching accounted for >50 relevant patient characteristics; all analyses were conducted separately in two databases and results were pooled using the DerSimonian and Laird random effects model with inverse variance weights. * In the as-treated approach, the follow-up was stopped if patients either filled a prescription for a drug in the other exposure group or discontinued the index exposure. ** In the ITT analysis, patients were followed in their index exposure group regardless of treatment change or discontinuation for up to 365 days

that elevated expression of TNF-α or IL-1β is significantly associated with increased risk of CAD[28]. Thus, we sought to determine whether hydroxychloroquine has direct anti-inflammatory effects on endothelial cells via these pathways as a potential beneficial mechanism in CAD.

We pretreated human aortic endothelial cells with 10–50 μM hydroxychloroquine and monitored the expression of *VCAM1* and *IL1B* genes in the presence and absence of the cytokine TNF-α. TNF-α (5 ng/ml) caused a robust increase in the expression of *VCAM1* and *IL1B*, and this pro-inflammatory effect was

significantly attenuated by all of the doses of hydroxychloroquine tested (Fig. 4b). Similarly, hydroxychloroquine decreased inflammatory responses to 10 and 20 ng/ml TNF-α, as demonstrated by its attenuation of TNF-α-mediated VCAM-1 and IL-1β protein upregulation (Fig. 4c).

Patients with rheumatoid arthritis are reported to have increased endothelial dysfunction[29] that correlates with cardiovascular disease risk[30]. Therefore, we next tested whether hydroxychloroquine altered TNF-α-induced suppression of *NOS3* expression[31], a known marker of endothelial (dys)

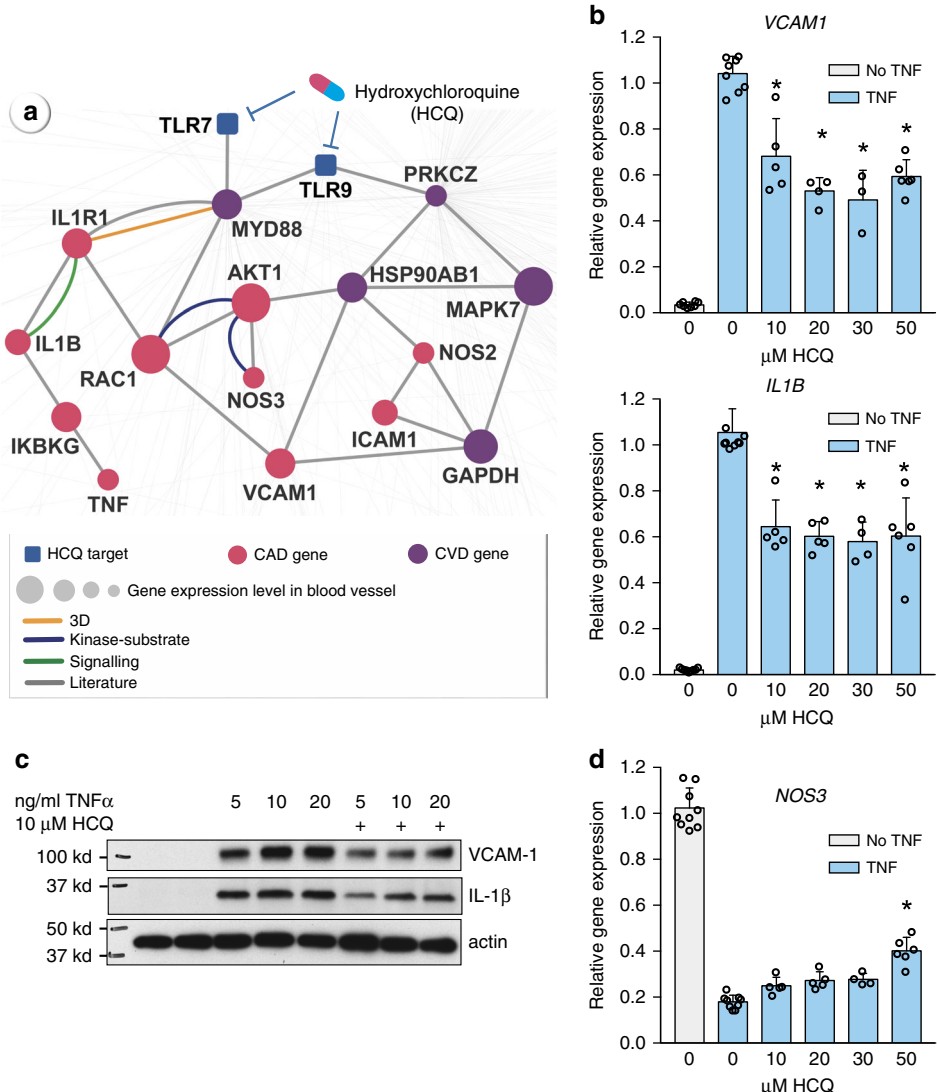

**Fig. 4** Experimental validation of hydroxychloroquine's likely mechanism-of-action in coronary artery disease (CAD). **a** A highlighted subnetwork shows the inferred mechanism-of-action for hydroxychloroquine's protective effect in CAD by network analysis. A network analysis was designed to meet four criteria: (1) the shortest paths from the known drug targets (TLR7 and TLR9) in the human protein–protein interaction network; (2) the blood vessel-specific gene expression level based on RNA-seq data from Genotype-Tissue Expression database; (3) known CAD or cardiovascular disease (CVD) gene products (proteins); and (4) literature-reported in vitro and in vivo evidence. There are three proposed mechanisms: (i) ERK5 (encoded by *MAPK7*) activation prevents endothelial inflammation via inhibition of cell adhesion molecule expression (VCAM-1 and ICAM-1), (ii) suppression of pro-inflammatory cytokines (TNF-α and IL-1β), and (iii) improvement in endothelial dysfunction via enhanced nitric oxide production by endothelial nitric oxide synthase (NOS3). The node size scales show the blood vessel-specific expression level based on RNA-seq data from Genotype-Tissue Expression database (Methods section). **b**, **d** Endothelial cells were pretreated with various concentrations of hydroxychloroquine (HCQ, 10–50 μM) for 1 h prior to 24 h incubation with 5 ng/ml TNF-α. qRT-PCR was used to monitor gene expression of inflammatory genes (**b**) *VCAM1* and *IL1B*; and (**d**) *NOS3*. Expression of the β-actin gene was used as an internal standard. VCAM1: no HCQ, no TNF, $n = 8$; TNF; $n = 8$; TNF+ 10 μM HCQ, $n = 5$; TNF+ 20 μM HCQ, $n = 4$; TNF+ 30 μM HCQ, $n = 3$; TNF+ 50 μM HCQ, $n = 6$. IL-1β and NOS3: no HCQ, no TNF, $n = 9$; TNF; $n = 9$; TNF+ 10 μM HCQ, $n = 5$; TNF+ 20 μM HCQ, $n = 5$; TNF+ 30 μM HCQ, $n = 4$; TNF+ 50 μM HCQ, $n = 6$. Error bars are standard deviations. *significantly different from TNF-α with no HCQ, $p < 0.05$ as determined by post-hoc testing using the Student's Newman–Keuls test. **c** Western blot of VCAM-1, IL-1β, and actin. Endothelial cells were pretreated with 10 μM hydroxychloroquine for 1 h prior to 24 h incubation with 5, 10 or 20 ng/ml TNF-α. Each condition was tested six times; shown are representative blots

function. *NOS3* encodes the endothelial nitric oxide synthase enzyme, which, via its synthesis of nitric oxide, regulates vascular tone, impairs platelet activation, and impairs adhesion molecule expression contributing to an anti-inflammatory (and anti-atherogenic) phenotype. TNF-α significantly suppressed *NOS3* expression, and 50 μM hydroxychloroquine significantly attenuated (reversed) this suppression (Fig. 4d). Taken together, network proximity analysis of the human interactome not only identified a novel protective effect of

hydroxychloroquine in CAD, but also offered testable hypotheses by which to elucidate the molecular mechanism(s) of its protective effect.

Although there may be additional pathways for the beneficial actions of hydroxychloroquine that are outside of the blood-vessel-specific protein–protein interaction network, these experimental findings suggest that hydroxychloroquine has a protective, anti-inflammatory effect on endothelial cells, consistent with its potential beneficial effect in CAD.

## Discussion

We have demonstrated that an integrated, mechanism-based human protein–protein interactome strategy can successfully uncover novel drug-disease indications, undesirable side effects, and potential mechanisms for these actions of approved drugs, addressing a crucial issue in drug development and patient care. We showed that our network framework yielded over 70% accuracy for identification of well-known drug indications (Supplementary Fig. 2). Specifically, our network-prediction and pharmacoepidemiological analysis reveal that carbamazepine is associated with an increased risk of CAD compared with levetiracetam, which we are able to validate robustly in large-scale patient data (HR 1.56, 95% CI 1.12–2.18, Fig. 3a). Carbamazepine is a first-line widely used anticonvulsant for the treatment of epilepsy and pain associated with trigeminal neuralgia, and works via inhibition of sodium channel protein type 5 subunit alpha (SCN5A)[32] and ATP-sensitive potassium (KATP) channels[33]. Previous clinical studies have suggested that carbamazepine aggravates high-grade heart block[34] and is associated with various cardiovascular risk factors[35], consistent with our observations. Moreover, recent genetic studies have shown that mutations in SCN5A and KATP channel genes are associated with structural heart disease[36] and adverse cardiac events[37,38]. Thus, it is mechanistically feasible that inhibition of SCN5A and KATP channel activities by carbamazepine may be associated with the increased risk of CV diseases. Further studies will be needed to provide experimental and clinical validation of this conclusion.

Pharmacoepidemiologic analyses from the two patient databases revealed an inconsistent association of lithium vs. lamotrigine on the risk of stroke: a null result (HR 1.02, 95% CI 0.79–1.33, Supplementary Fig. 4) in the Truven MarketScan database and a 52% reduced risk of stroke (HR 0.48, 95% CI 0.25–0.93, Supplementary Fig. 5) in the Optum Clinformatics database. Recent studies have shown the potentially protective effect of lithium in stroke[39]. Thus, to explore the effect of lithium in stroke further, we examined the potential molecular mechanism of lithium in the CV system via network analysis and in vitro assays of lithium exposure in cultured human aortic endothelial cells (Supplementary Fig. 6). We found a subnetwork of stroke genes and genes up- or down-regulated by lithium (Supplementary Fig. 6a) that map to pathways involved in the production of nitric oxide, which not only has anti-thrombotic effects but also vascular and neural protective effects in the central nervous system; however, our subsequent analysis in human aortic endothelial cells suggested that lithium may attenuate activation of these protective pathways (Supplementary Figs. 6b–e). In vitro assay results are consistent with a recent study that maternal use of high-dose lithium during the first trimester is associated with an increased risk of cardiac malformation in the foetus[40]. Thus, our findings suggest that larger clinical trials and additional mechanistic studies may be necessary to clarify lithium's action in stroke prevention in a broad population or a well-defined subpopulation.

Although patients with rheumatoid arthritis on hydroxychloroquine had a lower risk of CAD than rheumatoid arthritis patients treated with leflunomide, the ability of hydroxychloroquine to improve outcomes in patients with other underlying risk factors for CAD is unclear. Nonetheless, several CVD and CAD proteins are found within the hydroxychloroquine subnetwork (Fig. 4a). Furthermore, hydroxychloroquine has anti-inflammatory properties[41,42], and inflammation is a known major contributor to CAD[43]. In a mouse model of atherosclerosis, hydroxychloroquine was found to have anti-atherogenic and vasculoprotective effects[44], suggesting its utility in preventing vascular remodeling. Herein, our in vitro assays reveal that hydroxychloroquine attenuates the pro-inflammatory cytokine-mediated activation of human aortic endothelial cells (Fig. 4b–d) by reducing the expression of adhesion molecules, decreasing the production of cytokines, and attenuating the suppression of endothelial nitric oxide synthase. Although additional mechanistic studies are necessary to confirm the beneficial effects of hydroxychloroquine on endothelial function in the context of CAD, the anti-inflammatory properties of hydroxychloroquine on other cell types is well known. In rheumatoid arthritis and systemic lupus erythematosus, hydroxychloroquine has been suggested to mediate its anti-inflammatory action by inhibiting the activation of TLR7 and TLR9 that reside in endosomal/lysosomal compartments[23]. Recent evidence suggests that internalization of TNF-α receptors and other plasma membrane receptors to endosomal compartments may be a necessary step in the activation of certain ligand-induced signaling pathways[45]. Thus, hydroxychloroquine, which accumulates in endosomes, may interfere with the inflammatory actions of multiple types of membrane receptors.

In support of an effect of hydroxychloroquine on endosomal signaling, assembly of NADPH oxidase 2 complexes in the endosome in response to pro-inflammatory stimuli was attenuated by hydroxychloroquine to reduce superoxide generation in monocytes[46]. Additionally, in a monocytic cell line, hydroxychloroquine attenuated TNF-α and IL6 expression in response to IL-1β and TNF-α stimulation, respectively[46]. Interestingly, a treatment trial (the OXI trial) has recently been initiated to assess the efficacy of hydroxychloroquine in preventing recurrent CV events in patients with myocardial infarction[47] owing to its anti-inflammatory effects as well as its additional biological actions[48]. The results of this ongoing trial may provide further insights into the cardioprotective actions of hydroxychloroquine in a subset of non-rheumatoid arthritis patients.

Our pharmacoepidemiologic method, relying on very large patient-level longitudinal data, has several advantages. First, we used two large population-based cohorts to validate the hypothesized associations, which allowed for statistically robust testing of small effect sizes in relatively small treatment subpopulations. Pharmacy dispensing data from insurance claims were used to define exposure to medications. This approach is generally considered to be more accurate than self-reported drug use or medical records[49]. We also applied a large number of covariates to account for confounding in our studies using the recommended approach of PS-matching for improving inference from large healthcare databases, which are increasingly recognized by regulators and payers as a vital source of information through which to understand the safety and effectiveness of medications used in routine care[50]. We conducted multiple sensitivity analyses to rule out chance findings and attempted to replicate our analyses in a second large database. However, there remain certain limitations of this approach. Insurance claims data are primarily collected for administrative purposes and do not contain detailed clinical information; therefore, residual confounding is possible despite high-dimensional covariate adjustment. We defined outcomes purely by using a claims-based definition; although, we used validated and specific codes, endpoints could not be adjudicated. Finally, our databases did not contain information on patient ethnicity, which is also a limitation. Replication of the associations identified in this study using databases that contain information on ethnicity is recommended in future studies to rule out treatment effect heterogeneity by ethnicity.

In summary, we demonstrated that an integration of molecular network-based approaches and state-of-the-art pharmacoepidemiologic methods can facilitate rational strategies for drug repurposing and the detection of side effects. Specifically, we observed that hydroxychloroquine was associated with 24% reduced risk of CAD compared with leflunomide using large-

scale patient data, effects that are supported by mechanistic in vitro data. In addition, carbamazepine was associated with a 56% increased risk of CAD compared with levetiracetam. We believe that the approach presented here, if broadly applied, would significantly catalyze innovation in drug discovery and development.

## Methods

**Building the human protein–protein interactome**. To build the comprehensive human protein–protein interactome as currently available, we assembled 15 commonly used databases with multiple types of experimental evidence and the in-house systematic human protein–protein interactome: (1) binary PPIs tested by high-throughput yeast-two-hybrid (Y2H) systems in which we combined binary PPIs tested from two publicly available high-quality Y2H datasets[19,51] and one dataset available from our website: http://ccsb.dana-farber.org/interactome-data.html; (2) kinase-substrate interactions from literature-derived low-throughput and high-throughput experiments from KinomeNetworkX[52], Human Protein Resource Database (HPRD)[53], PhosphoNetworks[54,55], PhosphositePlus[56], dbPTM 3.0[57], and Phospho.ELM[58]; (3) carefully literature-curated PPIs identified by affinity purification followed by mass spectrometry (AP-MS), and from literature-derived low-throughput experiments from BioGRID[59], PINA[60], HPRD[53], MINT[61], IntAct[62], and InnateDB[63]; (4) high-quality PPIs from three-dimensional (3D) protein structures reported in Instruct[64]; and (5) signaling networks from literature-derived low-throughput experiments as annotated in SignaLink2.0[65]. The genes were mapped to their Entrez ID based on the National Center for Biotechnology Information (NCBI) database[66] as well as their official gene symbols based on GeneCards (http://www.genecards.org/). Inferred data, such as evolutionary analysis, gene expression data, and metabolic associations, were excluded. The updated human interactome constructed in this way includes 243,603 protein–protein interactions (PPIs) (edges or links) connecting 16,677 unique proteins (nodes) (Supplementary Data 1), representing over 40% greater size compared to our previously utilized human interactome[6].

**Collection of human cardiovascular disease genes**. We began with ~50 types of CV events defined by Medical Subject Headings (MeSH) and Unified Medical Language System (UMLS) vocabularies[67]. For each CV event, we collected disease-associated genes from 8 commonly used data sources: The OMIM database (Online Mendelian Inheritance in Man)[68], The Comparative Toxicogenomics Database[69], HuGE Navigator[70], DisGeNET[71], ClinVar[72], GWAS Catalog[73], GWASdb[74], and PheWAS Catalog (phewas.mc.vanderbilt.edu)[75]. We annotated all protein-coding genes using gene Entrez ID, chromosomal location, and the official gene symbols from the NCBI database[66]. Here we selected CV events with at least 10 disease-associated genes in the human interactome, resulting in 23 types of CV events (Supplementary Table 1).

**Construction of drug-target network**. We assembled the physical drug-target interactions on FDA-approved drugs from 6 commonly used data sources, and defined a physical drug-target interaction using reported binding affinity data: inhibition constant/potency ($K_i$), dissociation constant ($K_d$), median effective concentration ($EC_{50}$), or median inhibitory concentration ($IC_{50}$) ≤10 µM. Drug-target interactions were acquired from the DrugBank database (v4.3)[76], the Therapeutic Target Database (TTD, v4.3.02)[77], and the PharmGKB database (30 December 2015)[78]. Specifically, bioactivity data of drug-target pairs were collected from three commonly used databases: ChEMBL (v20)[79], BindingDB (downloaded in December 2015)[80], and IUPHAR/BPS Guide to PHARMACOLOGY (downloaded in December 2015)[81]. After extracting the bioactivity data related to the drugs from the prepared bioactivity databases, only those items meeting the following four criteria were retained: (i) binding affinities, including $K_i$, $K_d$, $IC_{50}$, or $EC_{50}$ ≤10 µM; (ii) proteins can be represented by unique UniProt accession number; (iii) proteins are marked as reviewed in the UniProt database[82]; and (iv) proteins are from *Homo sapiens*.

**Description of network proximity**. Given $S$, the set of disease proteins, $T$, the set of drug targets, and $d(S,T)$, the closest distance measured by the average shortest path length between nodes $s$ and the nearest disease protein $t$ in the human protein–protein interactome is defined as: $d(S,T) = \frac{1}{\|T\|} \sum_{t \in T} \min_{s \in S} d(s,t)$. To

evaluate the significance of the network distance between a drug and a given disease, we constructed a reference distance distribution corresponding to the expected distance between two randomly selected groups of proteins of the same size and degree distribution as the original disease proteins and drug targets in the network. This procedure was repeated 1000 times. The mean $\bar{d}$ and s.d. ($\sigma_d$) of the reference distribution were used to calculate a z-score ($z_d$) by converting an observed (non-Euclidean) distance to a normalized distance.

**Pharmacoepidemiologic methodology**. We conducted observational cohort studies using two large US-based health insurance claims databases: (1) Truven

MarketScan (2003–2014), and (2) Optum Clinformatics (2004–2013). These data sources contain comprehensive longitudinal information on patient demographics, coded in-patient and out-patient diagnoses and procedures, and outpatient prescription dispensing for their enrollees. Use of the de-identified database was approved by the Institutional Review Board of Brigham and Women's Hospital, Boston, MA.

We identified patients 18 years or older who initiated treatment with the drug of interest after 180 days of continuous enrollment[83]. The date on which this new prescription was filled was defined as the index date. We further applied study-specific exclusion criteria (summarized in Fig. 2) in the 180-day pre-index period to include homogeneous groups of patients in each comparison and focused on incident events. The follow-up began on the day after the index date. For the primary analysis, we used an as-treated follow-up approach in which the follow-up was stopped if patients either filled a prescription for a drug in the other exposure group or discontinued the index exposure. Discontinuation was defined as no record of a subsequent prescription of the index medication for 60 days after accounting for the days' supply of exposure provided by the most recent prescription. We varied the follow-up approach to evaluate the robustness of our results in three sensitivity analyses. First, we did not attribute the outcome occurring in the first 60-days post-index to the index treatment to avoid the possibility of unmeasured baseline confounding. Second, we truncated the follow-up to a maximum of 365-days to limit the potential for time-varying confounding. Finally, we conducted an intention-to-treat (ITT) equivalent analysis in which patients were followed in their index exposure group regardless of treatment change or discontinuation for up to 365 days. In all of the approaches, the follow-up was truncated at the first outcome occurrence, health insurance disenrollment, death, or the most recent date of data availability.

The outcome of CAD was identified as a composite endpoint of hospitalization for myocardial infarction as the primary discharge diagnosis or a coronary revascularization procedure. The ICD-9 codes and CPT codes used to identify these outcomes have been found to have >90% positive predictive value (PPV) in administrative claims databases[84,85]. The outcome of stroke was identified using hospitalization claims where ischemic stroke or transient ischemic attack was recorded as the primary discharge diagnosis. The ICD-9 codes used to identify this outcomes have been found to have 96% positive predictive value (PPV) in administrative claims databases[86].

We identified the large number of covariates, which were measured in the 180-day baseline period preceding each patient's index date, in each of the four studies to account for confounding. These variables were specifically selected to address clinical scenarios evaluated in each study. For example, in the study of inflammatory bowel disease (IBD) treatments (mesalamine vs. azathioprine), we measured and accounted for IBD severity-related variables, such as diagnosis for active fistula formation or internal penetrating disease, obstructing or stricturing disease, and intra-abdominal surgical procedures. Additionally, patient demographics (age and gender), risk factors for cardiovascular diseases (e.g., hypertension, hyperlipidemia, diabetes, cardiovascular medication use), and markers of contact with the healthcare system (e.g., number of emergency department visits, number of distinct prescription medications used) were measured in all four studies. Please refer to Supplementary Tables 2–5 for a full list of covariates included in each study.

We used propensity score (PS) methods to account for potential confounding[87]. PSs were defined as the predicted probability of receiving the treatment of interest (vs. the comparator) conditional upon patients' covariate constellations and were calculated using multivariable logistic regression models, including the covariates described above as independent variables. Initiators of each exposure of interest were matched to initiators of the reference exposure based on their PS in 1:1 ratio using a nearest-neighbor algorithm within a caliper of 0.05 on the probability scale[88]. Cox-proportional hazards models were used to estimate the adjusted hazard ratios (HR) between the treatment of interest and the risk of outcome before and after PS-matching. All analyses were conducted separately in the two data sources to avoid any potential effect of differential measurement of study variables across the data sources on the comparative estimates. The results were presented after pooling estimates from the two databases using the DerSimonian and Laird random effects model with inverse variance weights[89]. To address the possibility of population-overlap, we corrected the variance of our pooled hazard ratios assuming 20% overlap between the two databases as follows:

$$\widehat{\sigma^2}_{corrected} = \sum_{i=1}^{2} w_i^2 \widehat{\sigma^2}_i + w_1 w_2 \frac{n_1 \widehat{\sigma^2}_1 + n_2 \widehat{\sigma^2}_2}{n_1 + n_2} p_{overlap},$$

$\widehat{\sigma^2}_{corrected}$ = corrected variance,
$w_i$ = inverse variance weight for database $i$,
$\widehat{\sigma^2}_i$ = variance of the estimate from database $i$,
$n_i$ = sample size of the study in database $i$,
$p_{overlap}$ = 0.2.

All statistical analyses were conducted on the Aetion Platform version 2.1.2 using R (version 3.1.2), which has been validated against the FDA Sentinel system and randomized control trials[20].

**Tissue-specific subnetwork analysis**. We downloaded the RNA-seq data (RPKM value) of 32 tissues from GTEx V6 release (accessed on 01 April 2016,

https://gtexportal.org/home/). For each tissue (e.g., blood vessel), we regarded those genes with RPKM ≥1 in >80% of samples as tissue-expressed genes and the remaining genes as tissue-unexpressed. To quantify the expression significance of tissue-expressed gene $i$ in tissue $t$, we calculated the average expression $\langle E(i) \rangle$ and the standard deviation $\delta_E(i)$ of a gene's expression across all considered tissues[90]. The significance of gene expression in tissue $t$ is defined as $z_E(i, t) = (E(i, t) - \langle E(i) \rangle) / \delta_E(i)$. For stroke and CAD, we built a blood vessel-specific protein–protein interaction network by comparing genome-wide expression profiles of blood vessels to 31 other different tissues from GTEx.

In in vitro assays, human aortic endothelial cells (Lonza) were passaged in EGM-2 (Lonza) with the addition of hydroxychloroquine (Fig. 4) or lithium chloride (Supplementary Fig. 6) at the doses and times indicated. To assess VEGF-mediated activation of Akt/GSK/eNOS signaling, cells were cultured 24 h in EBM-2 with 0.1% fetal bovine serum in the presence of absence of lithium chloride prior to the addition of VEGF at 50 ng/ml for the times indicated (Supplementary Fig. 6). One hour after cells were exposed to hydroxychloroquine (10–50 μM), TNF-α (5–20 ng/ml) was added to the media. Cells were collected 24 h following TNF-α addition for RNA or protein analysis (Fig. 4).

RNA was collected from cells with the RNeasy kit (Qiagen) using the optional DNase I digestion. cDNA was synthesized from 0.5 μg of RNA using oligo dT primers and the Advantage RT-for-PCR kit (Clontech). Relative RNA levels were measured by quantitative RT-PCR method using the $\Delta\Delta C_t$ method of analysis. β-Actin was used as the endogenous control. The following TaqMan probes (Thermo Fisher) were used for gene expression analysis: *VCAM1*, Hs00365485_m1; *IL1B*, Hs01555410_m1; *NOS3*, Hs01574659_m1 and *ACTB*, Hs99999903_m1.

Radioimmunoprecipitation assay (RIPA) lysis buffer was supplemented with protease and phosphatase inhibitors (Calbiochem) and used to collect cell extracts. Cell lysates were separated on 4–15% polyacrylamide gradient gels (Biorad), and transferred to polyvinylidene fluoride (PVDF) membranes. Antibodies were obtained from Cell Signaling. VCAM-1 was detected by western blotting using an sc-8304 antibody (Santa Cruz) at a 1:4000 dilution; IL-1β was detected using a 1:1000 dilution of antibody #12703 (Cell Signaling); and actin was detected using a 1:4000 dilution of antibody #4970 (Cell Signaling). A secondary anti-rabbit-HRP antibody (Cell Signaling, #7074) was used at 1:2000 together with the ECL western blotting detection reagents from GE Healthcare. Blots were exposed to X-ray film, and the Biorad ChemiDoc Touch Imaging system was used to generate images. For western blot experiments designed to analyze the effects of hydroxychloroquine, each condition was tested in 6 independent experiments. Uncropped scans of the blots used in Fig. 4c are included in Supplementary Fig. 7.

**Code availability**. The toolbox package for the network proximity calculation can be downloaded at github.com/emreg00/toolbox.

**Data availability**. The human publicly available protein–protein interactome used in this study is freely available as a supplement to this manuscript (Supplementary Data 1). The unpublished binary human protein–protein interactions can be accessed at http://ccsb.dana-farber.org/interactome-data.html. The global predicted $z$-scores for 984 FDA-approved drugs and 23 types of cardiovascular events (diseases) via the network proximity approach are freely available in Supplementary Data 2. All other relevant data are available from the authors.

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

## Acknowledgements

This work was supported by the NIH grants P50-HG004233, U01-HG001715, and U01-HG007690 from NHGRI, P50-GM107618 from NIGMS and PO1-HL083069, R37-

HL061795, RC2-HL101543, U01-HL108630, RC4-HL106373, and K99HL138272 from NHLBI, and ME-1303–5638 from PCORI.

## Author contributions

J.L., A.-L.B., and S.S. conceived the study. F.C., R.J.D., D.E.H., performed all experiments and analysis. R.W. performed data analysis. F.C., R.J.D., D.E.H., S.S., A.-L.B., and J.L. wrote the manuscript.

## Additional information

**Competing interests:** A.-L.B. and J.L. are co-founders of Scipher, a startup that uses network concepts to explore human disease. S.S. is consultant to Aetion, Inc., a software manufacturer in which he also owns equity. The remaining authors declare no competing interests.

