## [Peer Review File · Nature Communications]

Reviewers' comments:

Reviewer #1 (Remarks to the Author):

This is a very interesting paper that uses an integrated, "systems biology" approach to discover drug-disease interactions. The authors assess cardiovascular disease associations for over 700 approved drugs. They then select four network-predicted associations and take a deeper dive with two such associations, establishing an association with coronary artery disease (CAD) with one drug (carbamazepine), while "protection" against CAD with another drug (hydroxychloroquine). They then do basic experiments to suggest biological plausibility for these findings.

The novelty of this paper is the general approach of the authors and the use of "network-based drug-disease proximity" to elucidate novel safety signals and drug-disease interactions. While one could always ask for "another experiment" (especially with the basic in vitro experiments) to further demonstrate the validity network hypotheses, I believe the novelty lies in the approach of the authors. For this reason, I believe this manuscript will be interesting to the scientific community and the readers of the journal.

I only have a few suggestions:

How do we know that hydroxychloroquine is actually protective against CAD and not merely less harmful than leflunomide? This concept is true with every such comparison that the authors make.

How does one take into account the background disease contributions to the CAD in each setting. RA (which hydroxychloroquine and leflunomide are used to treat) carries its own risk of CAD.

The experiments done for figure 4 are rather preliminary and are not truly proof of concept experiments. However, despite this shortcoming, I think the novelty lies in the authors' approach. In this regard, the authors should "tone down" the utility of their approach in terms identifying mechanisms of toxicity (or protection).

Reviewer #2 (Remarks to the Author):

The authors describe a new network-based method for predicting novel drug-disease associations, based on measures of network proximity. They statistically validated two of these associations using EHR data and propensity score matching. They then further validated one of these associations in vitro. The study provides a compelling methodology for computational drug repurposing.

It would be interesting to see these methods applied to multiple disease classes, cell/tissue types, and claims datasets (eg., FAERS, if the demographic data are sufficient).

I recommend this manuscript for acceptance, given the authors are able to address the few suggestions I have listed below:

- In lines 70-71, you touch on the idea of underrepresentation of relevant populations, but do not discuss or investigate this in the context of your study. What is known about the ethnicity content of your EHR data sources for the pharmacoepidemiology portion of the study? How might future studies improve on health disparities due to things like underrepresented ethnicities or other populations? I can't tell if important demographics like this were used in PSM; see the last bullet

point below.

- Given that the study was limited to cardiovascular drugs/diseases, this should probably be emphasized in the title and/or abstract out of caution that these methods may not generalize to other drug/disease contexts.
- Watch capitalization of $d(s,t)/d(S,T)$ in pg. 6, lines 119 and 120.
- Is the scale of the edge thickness legend in Figure 1 accurate? I can't see any edges in the network that come anywhere close to the thickness corresponding to -11.0 network proximity as indicated by that legend.
- Supplementary Tables S2-S6 seem to be missing, although their legends are present in the supplementary materials file. Since these provide the covariates used in PSM, they are important to include.

Reviewer #3 (Remarks to the Author):

The authors present a method for drug-repurposing based on network construction. The method rely on the construction of a network that integrates Protein-protein interactions (PPI), disease-gene and drug-target associations. The developed pipeline is very accurate and the authors integrated several data sources.

To extract meaningful ways of drug repurposing, the authors applied a strategy they recently developed that computes a proximity score for each association based on the distance between disease proteins and target proteins.

The methods applied are not novel, however the novelty of the paper rely on the data sources used and how they have been integrated, plus the validation approach is novel.

In fact, the authors filtered meaningful target associations based on the predicted network proximity scores, the exclusion of known adverse events, the availability of patient data for evaluation, availability of a comparator treatment, the fidelity which the predicted disease is recorded in insurance claim database. Very interesting they included pharmacoepidemiologic data using two health insurance databases containing longitudinal information on patients and used a propensity score method accounting form potential confounding to extract potential treatments to repurpose.

Taken together all these factors led to a robust and accurate method.

The authors applied the methodology only to coronary artery diseases (CDA). The authors identified four candidates, and in vitro tested a candidate. It would be nice to see how it performs also to other major disease, e.g. tumors, diabetes, etc. Otherwise I would suggest to adapt the title to CDA.

I also would like to see how the results vary by computing the network proximity scores by using other distance measures.

Minor:

Figure 1. Some names on the disease nodes are missing.

Re: Ms. # NCOMMS-18-05730A

Responses to Reviewer #1

This is a very interesting paper that uses an integrated, “systems biology” approach to discover drug-disease interactions. The authors assess cardiovascular disease associations for over 700 approved drugs. They then select four network-predicted associations and take a deeper dive with two such associations, establishing an association with coronary artery disease (CAD) with one drug (carbamazepine), while “protection” against CAD with another drug (hydroxychloroquine). They then do basic experiments to suggest biological plausibility for these findings. The novelty of this paper is the general approach of the authors and the use of “network-based drug-disease proximity” to elucidate novel safety signals and drug-disease interactions. While one could always ask for “another experiment” (especially with the basic in vitro experiments) to further demonstrate the validity network hypotheses, I believe the novelty lies in the approach of the authors. For this reason, I believe this manuscript will be interesting to the scientific community and the readers of the journal.

Response: We thank the reviewer for the positive comments regarding our study.

1. How do we know that hydroxychloroquine is actually protective against CAD and not merely less harmful than leflunomide? This concept is true with every such comparison that the authors make.

Response: We thank the reviewer for this comment. To address this point, we had required that the control drug in each comparison (leflunomide, levetiracetam, lamotrigine, and azathioprine) have z-scores less than -2 in order to demonstrate that an association between the selected control drug and the outcome of interest is unlikely to exist. Moreover, we also incorporated clinical insight into the selection of control drugs, only choosing those that had no known association with

the outcome of interest. The detailed explanation had been added to pages 7 and 8 of the revised manuscript.

2. How does one take into account the background disease contributions to the CAD in each setting. RA (which hydroxychloroquine and leflunomide are used to treat) carries its own risk of CAD.

Response: We thank the reviewer for this constructive comment. We agree with the reviewer that the disease itself (e.g., rheumatoid arthritis [RA]) is a potential risk factor for the outcome of interest. To account for this fact, we designed our study in a comparative manner. Comparing the outcomes between users of the drug of interest (e.g., hydroxychloroquine, a non-biologic disease modifying drug indicated for early RA) and users of the control drug that is used for a similar indication (e.g., leflunomide, also a non-biologic disease modifying drug indicated for early RA) ensures that the effect of the underlying disease (e.g., RA) is homogeneous across the two treatment groups. Focusing on relative measures of associations (ratios and differences) with such a study design as we did, one can isolate effects of the medication from effects of the underlying indication on a particular outcome of interest.

3. The experiments done for figure 4 are rather preliminary and are not truly proof of concept experiments. However, despite this shortcoming, I think the novelty lies in the authors' approach. In this regard, the authors should "tone down" the utility of their approach in terms identifying mechanisms of toxicity (or protection).

Response: We thank the reviewer for this comment. We agree with the reviewer that it is difficult to predict the overall utility of the approach in identifying mechanisms involved in toxicity or protection for a given drug-disease pair. In fact, the lithium data are illustrative of this point: unexpectedly, we found that lithium

attenuated the activation of protective pathways that were identified in a subnetwork of stroke genes (gene products) and lithium targets. Thus, our goal was to use network analysis as a means to identify connections between the drug and disease that can be used to generate testable hypotheses regarding *potential* mechanisms of actions. We have toned down any discussion of this approach to make it clear that this approach may point to potential (rather than definitive) drug mechanisms. Any findings *in vitro* require additional studies to show that these effects occur *in vivo*, as well. To clarify this point, we added to the Discussion the following: “Although additional mechanistic studies are necessary to confirm the beneficial effects of hydroxychloroquine on endothelial function in the context of CAD, the anti-inflammatory properties of hydroxychloroquine on other cell types is well known.” Furthermore, our studies in endothelial cells were prompted by the use of “the blood vessel-specific protein-protein interaction network to identify the overlapping pathways between hydroxychloroquine targets and CAD proteins.” In the Results, we write, “Although there may be additional pathways for the beneficial actions of hydroxychloroquine that are outside of the blood-vessel-specific protein-protein interaction network, these findings suggest that hydroxychloroquine has a protective, anti-inflammatory effect on endothelial cells, consistent with its potential beneficial effect in CAD.” As mentioned in the Discussion, the anti-atherogenic and vasculoprotective effects of hydroxychloroquine have been shown in a mouse model of atherosclerosis. Thus, our findings add new information about potential molecular mechanisms by which hydroxychloroquine may regulate vascular responses involved in atherogenesis. The detailed explanations have been added in pages 11-12 and 14 of the revised manuscript.

Responses to Reviewer #2

The authors describe a new network-based method for predicting novel drug-disease associations, based on measures of network proximity. They statistically validated two of these associations using EHR data and propensity score matching. They then further validated one of these associations in vitro. The study provides a compelling methodology for computational drug repurposing. It would be interesting to see these methods applied to multiple disease classes, cell/tissue types, and claims datasets (e.g., FAERS, if the demographic data are sufficient). I recommend this manuscript for acceptance, given the authors are able to address the few suggestions I have listed below.

Response: We thank the reviewer for these helpful comments.

1. In lines 70-71, you touch on the idea of underrepresentation of relevant populations, but do not discuss or investigate this in the context of your study. What is known about the ethnicity content of your EHR data sources for the pharmacoepidemiology portion of the study? How might future studies improve on health disparities due to things like underrepresented ethnicities or other populations? I can't tell if important demographics like this were used in PSM; see the last bullet point below.

Response: We thank the reviewer for the constructive comments. Unfortunately, the health insurance databases we used in this study do not contain information about patient ethnicity. Therefore, we were unable to account for ethnicity in our analysis. We have acknowledged this point as a limitation in the Discussion section, and have called for replication of these signals in future studies conducted in databases that contain information on ethnicity (i.e., Medicare) to rule out treatment effect heterogeneity by ethnicity.

2. Given that the study was limited to cardiovascular drugs/diseases, this should

probably be emphasized in the title and/or abstract out of caution that these methods may not generalize to other drug/disease contexts.

Response: We thank the reviewer for this suggestion. To address this concern, we have added statements that our network proximity approach can be generalized to other drug/disease pairs on page 4 of the revised manuscript, emphasizing that this analysis of cardiovascular disorders and drugs is an illustrative proof-of-concept of the general approach.

3. Watch capitalization of $d(s,t)/d(S,T)$ in pg. 6, lines 119 and 120.

Response: We thank the reviewer for this comment. The $d(s,t)$ denotes the average shortest path length between nodes, s (drug targets), and the nearest disease protein, t , in the human protein-protein interactome. The $d(S,T)$ denotes the closest distance between a drug (S) and a disease (T) measured by the average shortest path length. We have added this detailed explanation on page 6 of the revised manuscript.

4. Is the scale of the edge thickness legend in Figure 1 accurate? I can't see any edges in the network that come anywhere close to the thickness corresponding to -11.0 network proximity as indicated by that legend.

Response: We thank the reviewer for this comment. We have re-normalized the scale of edges in the revised Figure 1.

5. Supplementary Tables S2-S6 seem to be missing, although their legends are present in the supplementary materials file. Since these provide the covariates used in PSM, they are important to include.

Response: We thank the reviewer for this comment, and apologize for this submission problem. We have attached Supplementary Tables S2-S6 in this

re-submission.

Responses to Reviewer #3

The authors present a method for drug-repurposing based on network construction. The method relies on the construction of a network that integrates Protein-protein interactions (PPI), disease-gene and drug-target associations. The developed pipeline is very accurate and the authors integrated several data sources. To extract meaningful ways of drug repurposing, the authors applied a strategy they recently developed that computes a proximity score for each association based on the distance between disease proteins and target proteins. The methods applied are not novel, however the novelty of the paper rely on the data sources used and how they have been integrated, plus the validation approach is novel. In fact, the authors filtered meaningful target associations based on the predicted network proximity scores, the exclusion of known adverse events, the availability of patient data for evaluation, availability of a comparator treatment, the fidelity which the predicted disease is recorded in insurance claim database. Very interesting they included pharmacoepidemiologic data using two health insurance databases containing longitudinal information on patients and used a propensity score method accounting form potential confounding to extract potential treatments to repurpose. Taken together all these factors led to a robust and accurate method.

Response: We thank the reviewer for the positive comments regarding our study.

1. The authors applied the methodology only to coronary artery diseases (CDA). The authors identified four candidates, and in vitro tested a candidate. It would be nice to see how it performs also to other major disease, e.g. tumors, diabetes, etc. Otherwise I would suggest to adapt the title to CDA.

Response: We thank the reviewer for this comment. Our group is actively applying our integrative approach to other diseases, including another cardiovascular disease (i.e., heart failure with preserved ejection fraction) and a neurological disease (i.e., Parkinson disease). To allow for the detailed explanation of our methods and findings, we found that we could only include a limited number of results in the current manuscript. However, the reviewer's constructive comments will be used to guide our future studies.

2. I also would like to see how the results vary by computing the network proximity scores by using other distance measures.

Response: We thank the reviewer for this comment. In this study, we computed the network proximity scores using the closest measure. To respond to the reviewer's recommendation, we compared the closest distance-based network proximity scores with three other network distance measures: (1) shortest, (2) kernel, and (3) centre, as described in a previous study (Guney et al., *Nature Communications* 2016). Relying on 177 FDA-approved cardiovascular drugs and their known cardiovascular indications, we found that the closest measure (AUC = 74.2%) demonstrated the best performance compared to three other network distance measures: shortest (AUC = 52.4%), kernel (AUC = 61.2%), and centre (AUC = 49.3%). We have added the detailed results and description in Supplementary Figure 3 and pages 6 and 7 of the revised manuscript.

Reference

Guney, E., Menche, J., Vidal, M. & Barabasi, A. L. Network-based in silico drug efficacy screening. *Nat. Commun.* **7**, 10331 (2016).

3. Minor: Figure 1. Some names on the disease nodes are missing.

Response: We thank the reviewer for this comment. We have added all disease denotations in the revised Figure 1.

REVIEWERS' COMMENTS:

Reviewer #1 (Remarks to the Author):

The authors have adequately addressed my minor points in the rebuttal and revision of the manuscript.

Reviewer #2 (Remarks to the Author):

I'm happy with all of the revisions and responses to my suggestions.

Reviewer #3 (Remarks to the Author):

The authors well addressed the answers to the reviewers.

The paper has been improved and more emphasis is placed on the novelty of the approach rather than their discoveries and validations.

The approach used is novel and the author demonstrated its potential.